# Exploring the Multiscale Relationship between the Built Environment and the Metro-Oriented Dockless Bike-Sharing Usage

**DOI:** 10.3390/ijerph19042323

**Published:** 2022-02-17

**Authors:** Zhitao Li, Yuzhen Shang, Guanwei Zhao, Muzhuang Yang

**Affiliations:** 1School of Geography and Remote Sensing, Guangzhou University, Guangzhou 510006, China; 2112001071@e.gzhu.edu.cn (Z.L.); 2112001045@e.gzhu.edu.cn (Y.S.); ymz@gzhu.edu.cn (M.Y.); 2Institute of Land Resources and Coastal Zone, Guangzhou University, Guangzhou 510006, China

**Keywords:** bike-sharing, built environment, metro, geographically weighted regression, multiscale

## Abstract

Dockless bike-sharing systems have become one of the important transport methods for urban residents as they can effectively expand the metro’s service area. We applied the ordinary least square (OLS) model, the geographically weighted regression (GWR) model and the multiscale geographically weighted regression (MGWR) model to capture the spatial relationship between the urban built environment and the usage of bike-sharing connected to the metro. A case study in Beijing, China, was conducted. The empirical result demonstrates that the MGWR model can explain the varieties of spatial relationship more precisely than the OLS model and the GWR model. The result also shows that, among the proposed built environment factors, the integrated usage of bike-sharing and metro is mainly affected by the distance to central business district (CBD), the Hotels-Residences points of interest (POI) density, and the road density. It is noteworthy that the effect of population density on dockless bike-sharing usage is only significant at weekends. In addition, the effects of the built environment variables on dockless bike-sharing usage also vary across space. A common feature is that most of the built environment factors have a more obvious impact on the metro-oriented dockless bike-sharing usage in the eastern part of the study area. This finding can provide support for governments and urban planners to efficiently develop a bike-sharing-friendly built environment that promotes the integration of bike-sharing and metro.

## 1. Introduction

In recent years, use of dockless bike-sharing has become more popular due to its advantages in terms of health, convenience, flexibility, and so on. Since they do not require docking stations, the biggest advantage of dockless bike-sharing is flexibility and convenience compared with the traditional docked bike-sharing system. Previous studies show that the dockless bike-sharing system changes the daily travel mode of urban residents constantly, especially providing an effective solution to the “first and last mile” travel problems for residents [1,2,3]. In addition, cycling is proved to be beneficial to the environment as it can significantly reduce traffic’s energy consumption and carbon emissions [4,5,6]. Moreover, some studies have shown that cycling can help improve physical and mental health [7,8,9]. In short, dockless bike-sharing systems can bring many social, economic, and environmental benefits to cities [10].

Several challenges in urban transportation management were brought by the rapid spread of bike-sharing, undoubtedly. One of the main issues is to understand that how the urban built environment impacts the use of dockless bike-sharing. Previous studies have demonstrated that the built environment is a key factor that affects the usage of bike-sharing as it is associated with a location in urban settings [11,12,13,14]. However, most studies only explored the influencing factors of dockless bike-sharing ridership from the perspective of the entire study area, while the usages of dockless bike-sharing systems at the metro station level have rarely been studied. As we know, the bike-sharing is generally used for short-distance travel, particularly for the first and last mile travel problem of urban residents. Strengthening the connection between dockless bike-sharing and metro stations can effectively enhance the metro’s service area, thus promoting the realization of the transit-oriented development (TOD) strategy. Therefore, it is crucial to understand what built environment factors affect the bike-sharing ridership of a certain metro station and what the degrees of impact are.

The ordinary least square (OLS) model is usually applied as the global model to explore the relationship between dockless bike-sharing and the built environment [13,15]. However, the spatially varying effects of built environment factors are neglected by the studies using the OLS model. Some authors have proposed a geographically weighted regression model to capture the spatial relationship between built environment factors and bike-sharing usage [16,17,18]. Compared with the OLS model, the geographically weighted regression (GWR) model can illustrate the spatial varieties of built environment’s impact on dockless bike-sharing. It is regrettable that the traditional GWR model applies an average bandwidth to illustrate the impact scales of different built environment factors on dockless bike-sharing riding that ignores the diversity of impact scales of different built environment factors. However, ignoring the diverse spatial effects of different factors does not tally with the facts. The multiscale geographically weighted regression (MGWR) model is an improved version of the traditional GWR model proposed by Fotheringham et al. in 2017 [19]. Recent studies demonstrated that the MGWR model can produce more reliable estimation results than the GWR model, providing the diverse impact bandwidth of each factor [20,21]. However, researches on the multi-scale spatial relationship between the built environment and dockless bike-sharing usage connected to metros are still limited.

Thus, to fill this gap, this study is designed to investigate the spatial relationship between metro-oriented dockless bike-sharing ridership and built environment by introducing the MGWR model. This study attempts to answer the question from two aspects. (1) What are the differences in the effect of built environment factors on the dockless bike-sharing ridership connected to metro in different periods of the week, such as weekdays and weekends? (2) How do the effects of the built environment factors on metro-oriented bike-sharing ridership vary across space? First, the daily access and egress bike-sharing ridership on weekdays and weekends were selected as the dependent variables, respectively. Eleven built environment factors based on “5Ds” theory, namely density, design, diversity, distance to transit, and destination accessibility [22], were selected as initial independent variables. Second, a stepwise regression model was applied to screen the independent variables which have significant impact on the dependent variables. Third, the spatial relationships between the built environment factors and the metro-oriented bike-sharing ridership were fitted using the GWR model and the MGWR model, respectively. Finally, the spatial variations in the effect of built environment factors on metro-oriented dockless bike-sharing ridership were analyzed according to the regression results of the MGWR model.

The rest of this paper is organized as follows. Section 2 reviews the literatures about the impact of built environment on bike-sharing usage. Section 3 introduces the study area and data. The methodology of study is described in Section 4, including determining the research units of the bike-sharing usage and the built environment, screening the independent variables, spatial non-stationarity test, models, and the model evaluation metrics. Section 5 presents the results of the regression models and discussion on the spatially varying effects of built environment factors on metro-oriented bike-sharing ridership. The findings and perspectives generated by this study are summarized in Section 6.

## 2. Literature Review

Over the last decade, numerous scholars have explored the relationship between bike-sharing travel behaviors and impact factors such as road density, weather, land-use, and social demographics. In order to focus on the issue of bike-sharing travel and urban built environment, we review a particular group of studies where a regression model was used to analyze the relationship. Comprehensive reviews about bike-sharing usage can be found in papers by Albuquerque et al., Vallez et al., and Eren and Uz [23,24,25].

Previous studies shown that the urban built environment, characterized in terms of population density, proximity, land use types, and so on, has a significant impact on bike-sharing usage. For example, Chen and Ye [11] explored the non-linear effects of built environment on bicycle sharing using the Mobike bike-sharing trajectory data in Chengdu, China, and found that population density and employment density were two important factors influencing bike-sharing ridership. Beside population and employment density, several scholars found that bike lanes, accessibility to public transit, and the points of interest (POI) density can promote bike usage effectively [26,27,28,29,30]. However, some inconsistent or even contrary results were also found. For example, Zhang et al. [31] claimed that accessibility to public transit has no obvious impact on bike usage in Zhongshan, China. In addition, research in Taipei and Tokyo has shown that population and employment density have no impact on bike usage [32]. Therefore, the effects of the built environment on bike-sharing usage still need to be explored further.

From the perspective of research method, global regression models were usually applied in the early studies, including the ordinary least square (OLS) regression model, the spatial lag model (SLM), the spatial error model (SEM), the linear mixed-effects model, the negative binomial regression model, and so on. For example, Liu et al. [13] applied the ordinary least square regression model to examine the effects of built environments on the usage of bike-sharing. Guo and He [12] employed a negative binomial regression model to examine the effects of the built environment on the integrated use of dockless bike-sharing and the metro. Faghih-Imani and Eluru [33] and Wu et al. [14] applied spatial regression models, including a spatial lag model (SLM) and a spatial error model (SEM), to reveal the relationship between the importance of bike stations and the built environment. Faghih-Imani et al. [34] applied a linear mixed-effects model to explore what factors influence the arrival and departure flows at the station level.

Although the above-mentioned regression model can capture the built environment factors influencing bike-sharing usage, the spatial non-stationarity of variables is neglected by them. Thus, some scholars applied the geographically weighted regression (GWR) model to analysis the local effect of built environment on bike-sharing usage. For example, Wu et al. [17] demonstrated that the goodness of fit in the GWR model is better than that of the OLS model by applying a OLS model and a GWR model to examine the global and local influences of the built environment on bike usage. Li et al. [35] analyzed how the built environment and social-demographic characteristics influence bike-sharing utilization with the OLS model and the GWR model, and found that the shared bikes mainly serve a certain area instead of the whole city. Wang et al. [36] applied the GWR model to explore the relationship between bike-sharing usage in metro station service areas and its determinants, including the passenger flow in stations, land use, bus lines, and road network characteristics. They found that the bike-sharing usage around metro stations is mainly affected by the passenger flow into and out of stations, land use, bus lines, and road network characteristics and the bike-sharing usage around metro stations has obvious partition characteristics. Bao et al. [37] employed K-means clustering method to classify the bikeshare stations into five categories, and then applied five separate GWR models for each station category and compared with the joint model of all station categories, indicating that the prediction performance of separate bikeshare ridership models was generally better than that of the joint model. Li et al. [10] used a GWR model to analyze the spatial variation in the relationship between bike-sharing transfer distance and explanatory variables in Shanghai, and showed that transfer distance was related to factors such as daily patronage of metro stations, daily patronage of bike-sharing, population density, parking lot density, distance to central business district (CBD) and bus stop density. In addition, some studies found significant correlations between bike-sharing usage and eye-level greenness and frequency of public transport use [16,18]. Moreover, Gao et al. [38] discussed the modifiable areal unit problem in the impact of built environment factors on the usage of bike-sharing, and claimed that the influence of most built environment factors, such as the distance to metro stations, the number of work places, the number of residence facilities, the number of recreational places, land-use mixture and population density, are sensitive to the spatial areal units.

In summary, several scholars have explored the factors influencing the usage of bike-sharing using the traditional GWR model and obtained some meaningful insights. However, studies applying the traditional GWR model ignore the differences between the impact scales (the bandwidth in the GWR model) of independent variables, and applies an average bandwidth to describe the impact ranges of independent variables. Undoubtedly, the results of the GWR model are biased. Previous studies demonstrated that the multiscale geographically weighted regression (MGWR) model is more reliable than the GWR model in dealing with the spatial relationship regression issue [20,21,39]. Therefore, in this paper, the MGWR model is applied to explore the multi-scale spatial relationship between the built environment and the usage of bike-sharing connected to the metro stations in Beijing, China. Our findings can provide invaluable insights into policy formation regarding metro-oriented dockless bike-sharing usage to guide urban built environment optimization in China and worldwide.

## 3. Study Area and Data Description

### 3.1. Study Area

The study area of this research is Beijing, China. Beijing is the capital of China, which lies in the northern part of the north China plain. Beijing is adjacent to Tianjin in the East and Hebei Province elsewhere (see Figure 1). The population of Beijing in 2017 was 21.94 million [40], with an area of 16,410 km^2^. The terrain is high in the northwest and low in the southeast. Due to the congestion on the ground in Beijing, the metro has become the main mode of transport for residents of the city, and the average daily traffic of the metro in Beijing exceeds 12 million passengers [41,42]. By the end of 2017, there were 19 metro lines and 288 metro stations in operation, with a total length of 574 km. Metro stations are concentrated within the fifth ring road with some lines extending into Fangshan District, Tongzhou District, Shunyi District, Daxing District and Changping District. The topography of the area covered by the metro is relatively flat, so the effect of the topography on riding can be ignored [43]. Meanwhile, approximately 700,000 dockless bicycle-sharing bikes have been deployed in Beijing, with about 11 million registered users (equivalent to half of Beijing’s resident population) and about 7 million rides per day [44]. Among the bike-sharing companies in Beijing, Mobike was the largest dockless bike-sharing company [45]. The distribution of metro lines, metro stations, and the fifth ring road are shown in Figure 1.

### 3.2. Data Description

The research data are composed of two parts:(1)The bike-sharing trip record data are obtained from the competition of Mobike Big Data Challenge 2017. The original data set includes more than 1.83 million Mobike bike-sharing trip records in Beijing from 10–16 May 2017. By querying the weather condition records of Beijing, we can see that the weather was sunny and the air quality was also good in Beijing during that period [46], Therefore, the bike-sharing trajectory data during this period can reflect the real usage of Mobike bicycle in Beijing comprehensively. Each trip record includes order id, user id, vehicle id, vehicle type, start time, origin location, and destination location. The coordinate system of the origin and destination location is the GCJ-02 coordinate system, which is the official Chinese geodetic datum formulated by the Chinese State Bureau of Surveying and Mapping. The coordinates of origin location and destination location were encoded using the geohash, which is a method of encoding geographic coordinates to protect privacy information [47]. The principle of geohash is mapping all coordinates of a specific rectangular range to the same string, such that the longer the string, the higher the precision. Since the length of geohash code is 7 bits, the spatial resolution of the decoded bike-sharing ride original and destination point coordinates is approximately equal to 110 m * 150 m. The Manhattan distance formula is applied to calculate the riding distance for each trip order [48]. The calculation results show that more than 95% of the orders have a riding distance of 2000 m or less. As this study focuses on the characteristics of bike-sharing connected to the metro, in order to simplify the processing, the data with a riding distance of more than 2000 m are regarded as invalid data and eliminated.(2)Built environment data includes road network data, points of interest (POI) data and population density data. The road network data was obtained from the website of OpenStreetMap (https://www.openstreetmap.org/, accessed on 10 July 2017), including three road types: primary road, secondary road, and branch road. The POI data was obtained from Amap (https://www.amap.com, accessed on 7 July 2020), which includes 13 POI categories such as catering facilities, scenic spots, public service facilities, companies, shopping facilities, and transportation facilities. Each POI record includes name, type, location, latitude, and longitude of the specific location such that the coordinate system of location is also the GCJ-02 coordinate system. By cleaning the original POI data, 502,376 POI records were obtained for further analysis. The population density data were obtained from the WorldPop dataset (https://www.worldpop.org/project/categories, accessed on 1 December 2017), with a spatial resolution of 1 km × 1 km. Some studies have shown that the WorldPop dataset had the highest estimation accuracy in population datasets of China [49]. Finally, the coordinate systems of all data are unified to the WGS 1984 coordinate system.

## 4. Method

### 4.1. Determining the Research Units of the Bike-Sharing Usage and the Built Environment

In order to explore the spatial relationship between the built environment and the usage of bike-sharing connected to the metro in Beijing, the first step is to identify the bike-sharing trip used to connect to the metro station. Since the resolution of the bike-sharing data decoded by geohash is about 110 m × 150 m, a circular buffer zone with a radius of 150 m was created for the entrance/exit of each metro station (see Figure 2). The original and destination points that fall within the buffer zone can be considered as the trip records connected to the metro station. Then the original and destination points were extracted using the overlay tool in ArcGIS 10.2 software.

The built environment research unit was determined according to the riding distance of bike-sharing trips. As none of the pre-processed bike-sharing order data has a riding distance of more than 2 km, a reasonable hypothesis is that the built environment within a 2 km distance around a metro station is considered to have a significant impact on the bike-sharing riding connected to metro. A circular buffer zone with a radius of 2 km was created for each metro station (see Figure 3a). However, the density of metro stations in the central urban area was so high that the buffer zones overlapped with each other seriously. To deal with the overlap problem, the Thiessen-polygon method [50] was applied to create another buffer zone for the built environment around the metro stations (Figure 3b). Finally, the ultimate research unit of built environment was determined (see Figure 3b).

### 4.2. Screening the Independent Variables

According to previous studies, 11 initial indicators were selected to describe the built environment based on the “5Ds” theory, based on density, design, diversity, distance to transit, and destination accessibility [22]. However, the multicollinearity of independent variables, which is a common issue in regression analysis, can make the regression model biased or distorted. Therefore, the multicollinearity problem must be solved before fitting the regression model. Previous studies shown that the stepwise regression method can eliminate insignificant variables and variables that are highly correlated with other variables [51]. Thus, it can be used to screen the ultimate variables through dealing with the multicollinearity problem. In this study, the initial built environment variables were screened using the stepwise regression analysis tool of SPSS software (IBM SPSS Statistics, Chicago, IL, USA). According to previous studies, the variables with variance inflation factor (VIF) values greater than 10 should be excluded to eliminate the effects of multicollinearity [52].

### 4.3. Spatial Non-Stationarity Test

Spatial non-stationarity is a common issue of spatial regression analysis where the relationship or structure changes with geographic location. Previous studies have shown that the spatial heterogeneity of geographic environments will lead to spatial non-stationarity of relationships among variables. One of the basic assumptions of the ordinary least square (OLS) regression model is that the independent variables are independent of each other, which is manifested in spatial regression modeling as the residual terms of the model exhibiting random distribution characteristics. However, the assumptions of the OLS model are undermined by the possible existence of spatial autocorrelation or spatial dependence. Therefore, the existence of spatial non-stationarity is a prerequisite for applying spatial regression models. As mentioned earlier, the spatial non-stationarity of the regression model is tested by examining the degree of randomness of the model residual distribution. In this paper, the Moran’s I index is utilized for measuring of the model’s residual distribution [53]. The Moran’s I is a widely used metric that measures the spatial autocorrelation of variables. The expected values of Moran’s I usually range from −1 to 1. A value of Moran’s I closer to 1 indicates a higher degree of positive spatial autocorrelation. Similarly, a value of Moran’s I closer to −1 indicates a higher degree of negative spatial autocorrelation. If the values of variable were truly randomly dispersed, the Moran’s I value would be 0 (perfect randomness). The formula of Moran’ s I can be expressed as follows:(1)$$I=\frac{n}{S_{0}}\frac{\sum_{i=1}^{n}\sum_{j=1}^{n}w_{i,j}z_{i}z_{j}}{\sum_{i=1}^{n}z_{i}^{2}}$$
where $z_{i}$ represents the deviation between the attribute of i and its average value, $w_{i,j}$ represents the spatial weights between i and j. n represents the total number of the sample. $S_{0}$ represents the aggregation of all spatial weights. The formula of $S_{0}$ is as follows:(2)$$S_{0}=\overset{n}{\underset{i=1}{\sum }}\overset{n}{\underset{j=1}{\sum }}w_{i,j}$$
where $w_{i,j}$ represents the spatial weights between i and j. n represents the total number of the sample. The formula of $z_{i}$ is as follows:(3)$$z_{I}=\frac{I-E\left[I\right]}{\sqrt{V\left[I\right]}}$$
where E[I] represents the mean value. V[I] represents the variance. The formulas of E[I] and V[I] are listed as follows:(4)E[I]=−1/(n−1)
(5)$$V\left[I\right]=E\left[I^{2}\right]-E{\left[I\right]}^{2}$$

### 4.4. Models

#### 4.4.1. The Geographically Weighted Regression Model

According to the research of Anselin et al., the ordinary least square model on the global scale should be always the beginning of spatial regression analysis [54]. However, the global OLS model cannot clearly explain the spatial relationships between bike-sharing use and built environment variables. The geographically weighted regression model is an extended form of the global linear regression model, which can explore the heterogeneity of geospatial variables. It can be used to explore the non-stationarity of geospatial variables, which vary in relation to each other due to geographical location [55]. Previous studies have demonstrated that the geographically weighted regression (GWR) model is more advantageous in dealing with the non-stationarity of spatial relationships than the global regression models [56,57]. The GWR model can be expressed as follows:(6)$$y_{i}=\beta_{0}\left(u_{i},v_{i}\right)+\underset{k}{\sum }\beta_{k}\left(u_{i},v_{i}\right)x_{ik}+\varepsilon_{i}$$
where i represents research unit, $y_{i}$ represents the number of O or D points of i, $\left(u_{i},v_{i}\right)$ represents the location of i, $\beta_{0}$ represents the intercept, $\beta_{k}$ represents the coefficients, $x_{ik}$ represents the independent variable, and $\varepsilon_{i}$ represents the error term. The GWR model estimator can be expressed as follows:(7)$$\hat{\beta }\left(u_{i},v_{i}\right)={\left[X^{T}W\left(u_{i},v_{i}\right)X\right]}^{-1}X^{T}W\left(u_{i},v_{i}\right)Y$$
where X represents the matrix of independent variables, Y represents the vector of dependent variables, and $W\left(u_{i},v_{i}\right)$ represents the matrix of weights, associated with spatial locations. The weight matrix can be expressed as follows:(8)$$W\left(u_{i},v_{i}\right)=\left[\begin{array}{ccc} W_{1}\left(u_{i},v_{i}\right) & 0 & 0\\ 0 & \ddots & 0\\ 0 & 0 & W_{n}\left(u_{i},v_{i}\right) \end{array}\right]$$
where n represents the number of research units.

#### 4.4.2. The Multiscale Geographically Weighted Regression Model

Although the GWR model has been widely used for spatial non-stationarity issues, an obvious drawback is that it adopts a globally average bandwidth for all variables to describe the impact range of each variable. Undoubtedly, the estimation result of the GWR model may be biased due to ignoring the variety of impact scales among the variables. The multiscale geographically weighted regression (MGWR) model is an upgraded version of the GWR model, which was proposed by Fotheringham in 2017 [19]. Compared with the GWR Model, the MGWR model can obtain a more reliable result by providing an optimal bandwidth for each independent variable [58]. The MGWR model can be expressed as follows:(9)$$y_{i}=\underset{k}{\sum }\beta_{bwk}\left(u_{i},v_{i}\right)x_{ik}+\varepsilon_{i}$$
where $\beta_{bwk}$ represents the regression coefficient of the variable under different bandwidth conditions, $x_{ik}$ represents the independent variable, and $\varepsilon_{i}$ represents the error term.

### 4.5. Model Evaluation Metrics

In this study, several commonly used metrics are applied to compare the results of involved regression models, including the R-squared($R^{2}$), the Adjusted R-squared (*Adj*. $R^{2}$) and the AICc values [59]. $R^{2}$ is a classic metric that can measures the goodness of fit of a regression model. The closer the $R^{2}$ value is to 1, the better the fitting effect of the model. $R^{2}$ can be expressed as follows:(10)$$R^{2}=1-\frac{\sum {\left(y_{i}-{\hat{y}}_{i}\right)}^{2}}{\sum {\left(y_{i}-\bar{y}\right)}^{2}}$$
where $y_{i}$ represents the true value, ${\hat{y}}_{i}$ represents the fitted value, and $\bar{y}$ represents the average value. In the actual analysis, $R^{2}$ is usually replaced by *Adj*. $R^{2}$ which is a revised version of $R^{2}$ that removes the effect of the number of independent variables on $R^{2}$. The *Adj*. $R^{2}$ can be expressed as follows:(11)$$Adj.\;R^{2}=1-\left[\frac{\left(1-R^{2}\right)\left(n-1\right)}{\left(n-k-1\right)}\right]$$
where n represents sample size, and k represents the number of independent variables. In addition, the AICc value is also a popular indicator for evaluating models, and a model with a smaller AICc value has a better fitting result. AICc can be expressed as follows:(12)$$AICc=2nln\left(\sigma \right)+nln\left(2\pi \right)+n\left(\frac{n+tr\left(S\right)}{n-2-tr\left(S\right)}\right)$$
where n represents the number of observations, σ represents the standard deviation of the error term, and tr(S) represents the trace of the S-matrix of the regression model, as a function of bandwidth.

## 5. Results and Analysis

### 5.1. Building and Results of Regression Models

Based on previous research, we designed four types of regression models for bike-sharing daily usage in weekdays and weekends, respectively. To describe the dockless bike-sharing travel behaviors more precisely, four types of dependent variables, including the access-daily use on weekdays, the egress-daily use on weekdays, the access-daily use on weekends, and the egress-daily use on weekends, were fitted in our study. Then, 11 variables were selected to describe the urban built environment according to the “5Ds” theory, based on density, design, diversity, distance to transit, and destination accessibility [22]. Specifically, the variables for measuring density and diversity include the population density and the land-use mixture [60]. Meanwhile, the variables for measuring design include primary road density, secondary road density, and branch road density. Moreover, distance to central business district (CBD) and bus station density were selected as the indicators of destination accessibility and distance to transit, respectively. In addition, the original points of interest (POI) types were reclassified according to the needs of our particular case study. The specific reclassification operations include combining the Restaurants POI and the Shopping malls POI into a Restaurants-Shopping POI, combining the Companies POI and the Banks POI into a Companies-Banks POI, combining the Hotels POI and the Residences POI into a Hotels-Residences POI, and combining the Schools POI and the Sports facilities POI into a Schools-Sports POI. The density of four types of reclassified POI were chosen as the independent variables in this study. The initial dependent variables and independent variables of regression models are shown in Table 1.

Table 2 shows the results of the four stepwise regression models. It can be seen from Table 2 that the Distance to CBD, the Hotels-Residences POI density and the primary road density have a significant impact on both access-daily and egress-daily use on weekdays. In addition, the secondary road density and the branch road density are also significantly associated with access-daily use and egress-daily use, respectively, on weekdays. For weekends, the results demonstrate that both access- and egress-daily use are significantly affected by distance to CBD, Hotels-Residences POI density, primary road density, secondary road density, and population density. Since the variance inflation factor (VIF) values of all variables are less than 10, it can be assumed that the multicollinearity problem does not exist in the models constructed by stepwise regression.

### 5.2. The Fitting Results of Regression Models

The existence of spatial autocorrelation of variables is a prerequisite to applying the geographically weighted regression (GWR) Model. The spatial autocorrelation test results of independent variables are shown in Table 3 using the Moran’ s I.

As shown in Table 3, the Moran’ s I values for all variables are greater than 0.3 and their *p*-Values are less than 0.01. It can be inferred that the built environment variables involved in this model have significant positive spatial autocorrelation between each other. Therefore, the GWR series models are suitable to explain the spatial relationships by dealing with the spatial dependence of independent variables. The evaluation results of the ordinary least square (OLS) model, the GWR model and the multiscale geographically weighted regression (MGWR) model fitted for four types of dockless bike-sharing daily usage are shown in Table 4.

From Table 4, we can see that the $R^{2}$ and the adjusted $R^{2}$ of the GWR model and the MGWR model are both higher than the OLS model, indicating that the explanatory power of the GWR-based models has been significantly improved. According to the evaluation principle proposed by Fotheringham et al. [19], when the AICc value of a model decreases more than 3, it can be considered that there is a significant improvement in the goodness of fit of the model. Compared with the OLS model, the AICc values of GWR-based models decreased significantly, indicating a great improvement in the goodness of fit of the model. In addition, compared with the OLS model, the Moran’ s I values of the residuals of the GWR model and the MGWR model are both less than 0.06 and their *p*-Values are not significant at the 0.1 level, indicating that there is no significant spatial autocorrelation in the residuals of the GWR model and the MGWR model. Therefore, it can be inferred that the GWR model and the MGWR model both can deal with the spatial non-stationarity issue that cannot be resolved by the OLS model.

Among the two GWR-based models, the AICc value of the MGWR model is significantly lower than the GWR model on both weekdays and weekends (the reduction of AICc is much greater than 3). Moreover, the adjusted $R^{2}$ and the Moran’ s I value of the model residuals are both better than that of the GWR model, indicating that the fitting result of the MGWR model is more accurate and reliable. Due to the spatial heterogeneity of variables [61], the GWR-based model is more suitable to reveal the influence mechanisms of variables in space than the OLS model. From our results we can infer that the MGWR model can improve the performance of the GWR model by giving each variable a different bandwidth to replace the average bandwidth applied in the GRW model.

In order to analysis the differences of the variable bandwidths between the GWR model and the MGWR model, the variable bandwidths of two models were plotted as shown in Figure 4.

It can be seen from Figure 4 that the bandwidth of independent variables in the GWR model were fixed and relatively small. In contrast, the bandwidths of the independent variables in the MGWR model are more diverse. For access-daily use on weekdays, the bandwidths of Hotels-Residences POI density and primary road density are close to the total sample, indicating that the influence is similar across space with less heterogeneity. The secondary road density has a large bandwidth and the coefficients are smoother across space. The bandwidths of distance to CBD were approximately 20% of the sample size. Since the metro stations are concentrated in five urban districts, including Dongcheng District, Xicheng District, Chaoyang District, Haidian District and Fengtai District, the bandwidth of distance to CBD is close to the district scale on average, indicating that the coefficients do not vary much within each district. For egress-daily use on weekdays, the bandwidths of primary road density and branch road density are both close to the total sample size, indicating that they may affect the metro-oriented dockless bike-sharing riding at the global scale. In contrast, the impact scales of Hotels-Residences POI density and distance to CBD on bike-sharing riding are close to the district scales, and their coefficients varied drastically in space. For access-daily and egress-daily use on weekends, the Hotels-Residences POI density, primary road density, and secondary road density affect the bike-sharing riding at the global scale, while distance to CBD and population density affect the bike-sharing riding at district scale.

### 5.3. Discussion on the Spatially Varying Effects

In order to analyze the spatial heterogeneity of the impact of each built environment variable on the usage of bike-sharing connected to the metro, the local estimated coefficients of each variable in the MGWR model are shown in Figure 5, Figure 6, Figure 7, Figure 8 and Figure 9, respectively. As can be seen in Figure 5, the effects of distance to CBD on the usage of bike-sharing connected to the metro are roughly divided by the fifth ring road of Beijing. Within the fifth ring road, the distance to CBD is positively associated with bike-sharing usage, while outside the fifth ring road, they are negatively associated. It indicates that, within the fifth ring road, as the distance to the CBD increases, the more bike-sharing usage connected to the metro occurs, while outside the fifth ring road the opposite is true. The reason is probably that Beijing has adopted a strict transport control strategy within the fifth ring road. Therefore, the closer the area to the CBD, the more frequently trips by metro and another means of public transportation are generated. The obvious consequence is that the usage of dockless bike-sharing as the “first or last mile” of transport is significantly increased. In contrast, it can be inferred that residents away from the CBD are less likely to use bike-sharing to connect with the metro stations for their trips.

Figure 6 shows that Hotels-Residences POI density is positively associated with the usage of bike-sharing connected to the metro stations, which indicates that dockless bike-sharing is the main transportation means for people between residential areas and metro stations. Notably, for access-daily use on weekdays, the value of estimated coefficients for Hotels-Residences POI density varies from 0.470 to 0.525, and represents a spatial pattern such that the effect of Hotels-Residences POI density on the usage of bike-sharing gradually increases from northwest to southeast in the study area. For egress-daily use on weekdays, the effect of Hotels-Residences POI density is generally greater in the eastern part of the study area and Xicheng district, indicating that people who live in these areas are more likely to use bike-sharing as a feeder after taking the metro. For access-daily use on weekends, the effect of Hotels-Residences POI density on the usage of bike-sharing increases from western to eastern of the study area with the estimated coefficients varying from 0.383 to 0.409. For egress-daily use on weekends, the effect of Hotels-Residences POI density on the usage of bike-sharing increases from northwest to southeast with the value of estimated coefficients varying from 0.462 to 0.491.

Figure 7 shows the local estimated coefficients of primary road density in the MGWR model. It can be seen from Figure 7 that the absolute values of their coefficients are very small, indicating that the effect of primary road density on bike-sharing usage is extremely weak. In addition, it is worth noting that the primary road density has a negative effect on access-daily usage of bike-sharing and a positive effect on egress-daily usage of bike-sharing on both weekdays and weekends. It can be inferred that people who live in the areas with a high density of primary road are more likely to use bike-sharing as a feeder mode to the metro, while the opposite is true from the departure places close to the metro stations. A possible explanation is that most of the primary roads are the connections between districts. In the departure places where the main roads are dense, people may tend to drive private cars. 

Figure 8a shows the local estimated coefficients of branch road density in the MGWR model. It can be seen from Figure 8a that the branch road density is only positively associated with the egress-daily usage of bike-sharing on weekdays, indicating that the dense branch roads on weekdays will prompt people to choose bike-sharing as a feeder mode to metro. In addition, the effect of branch road density gradually increases from northwest to southeast in the study area with the estimated coefficients varying from 0.101 to 0.129. The local estimated coefficients of secondary road density in the MGWR model are presented in Figure 8b–d. For access-daily use on weekdays (see Figure 8b), the estimated coefficients vary from 0.015 to 0.054 in the northeastern of the study area, indicating that the secondary road density has a positive effect on the usage of bike-sharing, while the estimated coefficients in the remain areas ranging from −0.179 to −0.018, indicating that it has a negative effect in the rest of the study area. For both access-daily use and egress-daily use on weekends (see Figure 8c,d), the secondary road density is negatively associated with the usage of bike-sharing. A possible explanation is that the secondary roads are the main traffic arteries in the city which support the operation of various motor vehicles such as buses and private cars. However, due to the lack of dedicated bike lanes on these roads, the usage of dockless bike-sharing was reduced to a certain extent.

Figure 9 shows the local estimated coefficients of population density in the MGWR model. From Figure 9 we can see that there is a significant association between population density and bike-sharing usage on weekends; however, the association is not significant on weekdays. Moreover, from the central to the southeastern of the study area, the population density has a positive effect on the usage of bike-sharing, while a negative effect can be observed in Haidian district and Changping District. A possible explanation is that the purpose of cycling during weekdays is mainly for commuting, and the distribution of travel demand is more fixed, thus leading to an insignificant relationship between population density and dockless bike-sharing usage. On the contrary, as the travel behaviors of residents are more diverse and random on weekends, the dockless bike-sharing usage is affected by the population density. Especially in Haidian district and Changping district, there are a large number of colleges and universities with high population density, and metro entrances are usually set up near colleges and universities. Therefore, the bike-sharing usage connected to the metro occurs less frequently in these areas.

## 6. Conclusions and Prospect

Dockless bike-sharing, which can effectively expand the service area of rail transit, has become an important mode of transport for urban residents. This study explored the multiscale spatial relationship between the built environment and the usage of bike-sharing connected to metro stations by using several spatial regression models. The major findings from this study can be summarized as follows:The multicollinearity test result of the ordinary least square (OLS) model showed that among the initial urban built environment variables, the distance to central business district (CBD), the Hotels-Residences points of interest (POI) density, the primary road density, the secondary road density, the branch road density, and the population density have significant impact on the dockless bike-sharing riding connected to the metro stations. However, the spatial autocorrelation test results of the residual of OLS model proved the existence of spatial non-stationarity.The geographically weighted regression (GWR) model and multiscale geographically weighted regression (MGWR) model can both solve the problem of spatial non-stationarity compared with the OLS model. Since the MGWR model can identify the difference of effect scales between each variable, it is more reliable than the GWR model and the OLS model for analyzing the spatial relationship between the built environment and the usage of bike-sharing connected to the metro stations.Whether on weekdays or at weekends, the Hotels-Residences POI density and the branch road density have a positive effect on bike-sharing usage, while the secondary road density has a negative effect on bike-sharing usage. The population density only has a positive effect on bike-sharing usage on weekends. The primary road density has a negative effect on access-daily usage of bike-sharing and a positive effect on egress-daily usage of bike-sharing.The effects of the built environment variables on bike-sharing usage vary in space. The estimated coefficient of the distance to CBD varies most across space, while the estimated coefficient of the primary road density varies less. Within the fifth ring road, the distance to CBD has a positive effect on bike-sharing usage, while outside the fifth ring road, the effect is negative. In addition, most of the remaining variables have a greater effect on bike-sharing usage in the eastern part of the study area.

These findings can provide some implications for the better usage of dockless bike-sharing and metro systems. Firstly, the bike-sharing companies can optimize the configuration of bicycles depending on the built environment. This is because our study findings show that the effects of the built environment variables on bike-sharing usage vary in space. Then, as we found that the relationship between built environment and bike-sharing is different on weekdays and weekends, the dockless bike-sharing companies could dynamically adjust the supply of shared bikes around metro stations according to weekdays and weekends. Last but not least, due to the strong relationship between the built environment and bicycle sharing, the government could further improve the riding environment and infrastructure around the metro stations to promote the transfer between dockless bike-sharing and the metro.

This study attempts to introduce the MGWR model to explore the effect of the urban built environment on the usage of bike-sharing connected to metro stations, but there are still some limitations that one should consider for future studies. (1) The timing of the data is inconsistent. For example, the Mobike bike-sharing trip data in this study was obtained in 2017, while the POI data was obtained in 2020, which may affect the accuracy of model results. (2) The accuracy of the data needs to be improved. The coordinates of the bike-sharing trip data in this study were decoded by geohash with an accuracy of about 110 m * 150 m. Undoubtedly, there may be bias in identifying the bike-sharing trips connected to the metro station. (3) This research only focuses on the bike-sharing trip origin and destination, and the social demographic characteristics of residents were not involved. Considering the socio-economic characteristics of passengers for dockless bike-sharing would be meaningful work for future studies. (4) Topography, as an important element of the natural environment, may influence the configuration of metro lines in many cities. Therefore, the relationship between topography, metro stations, and bicycle sharing may need to be explored in future studies. (5) The results of this study have not been verified in other regions; hence, the generalizability of the results cannot be guaranteed. Comparison with other studies elsewhere needs to be developed in further studies.

## Figures and Tables

**Figure 1 ijerph-19-02323-f001:**
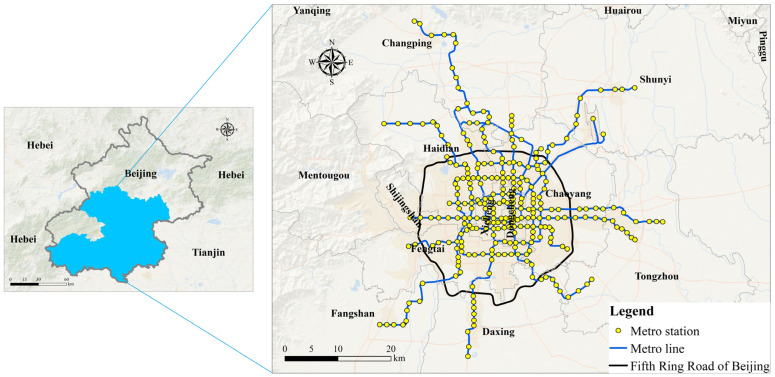
The location of Beijing, and the spatial distribution of metro stations, metro lines, and fifth ring road in the urban area of Beijing.

**Figure 2 ijerph-19-02323-f002:**
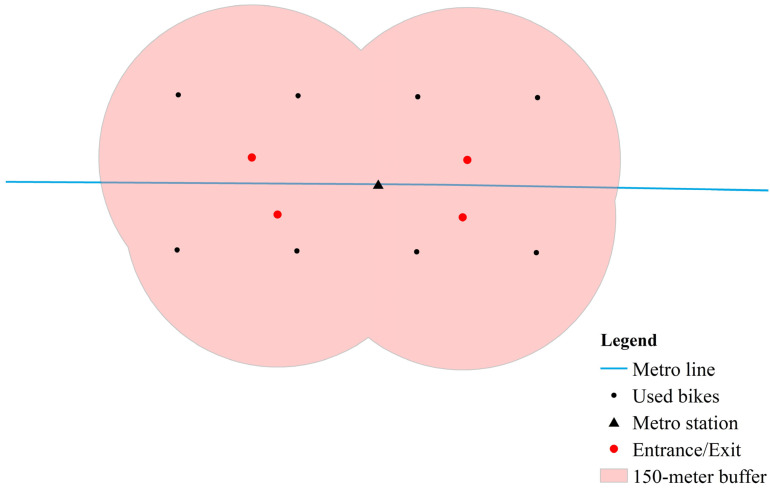
150-m buffer for Metro entrance/exit.

**Figure 3 ijerph-19-02323-f003:**
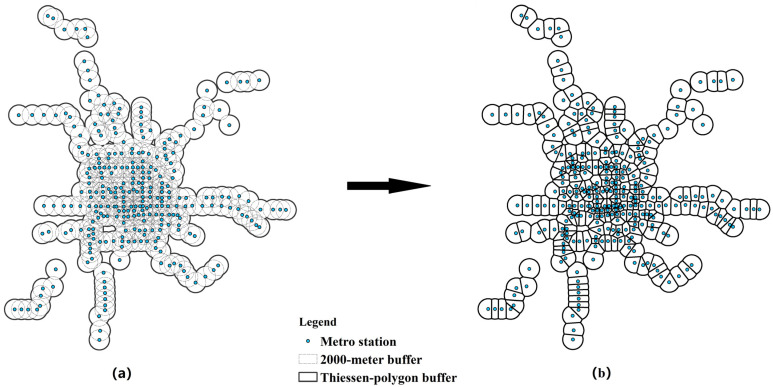
(**a**) 2000-m buffer (**b**) Thiessen-polygon buffer.

**Figure 4 ijerph-19-02323-f004:**
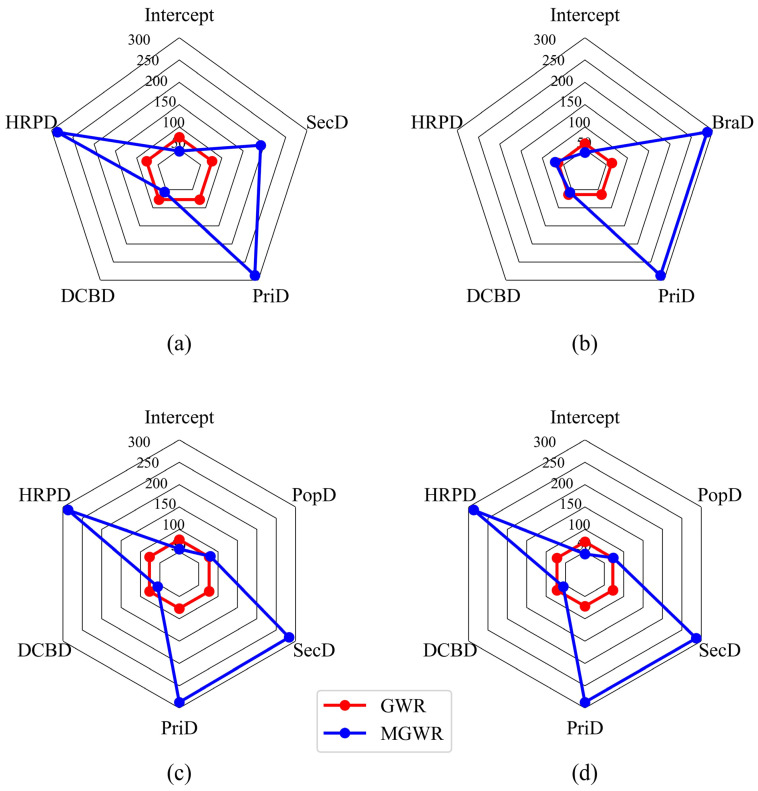
Comparison of bandwidths between GWR and MGWR models. (**a**) Access-daily use on weekdays; (**b**) egress-daily use on weekdays; (**c**) access-daily use on weekends; (**d**) egress-daily use on weekends.

**Figure 5 ijerph-19-02323-f005:**
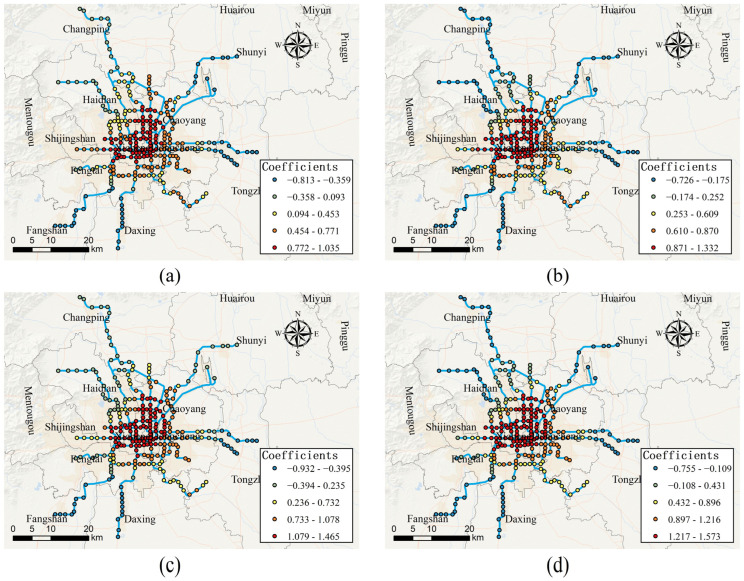
Spatially varying effects of distance to CBD. (**a**) Effect of distance to CBD on access-daily use on weekdays; (**b**) effect of distance to CBD on egress-daily use on weekdays; (**c**) effect of distance to CBD on access-daily use on weekends; (**d**) effect of distance to CBD on egress-daily use on weekends.

**Figure 6 ijerph-19-02323-f006:**
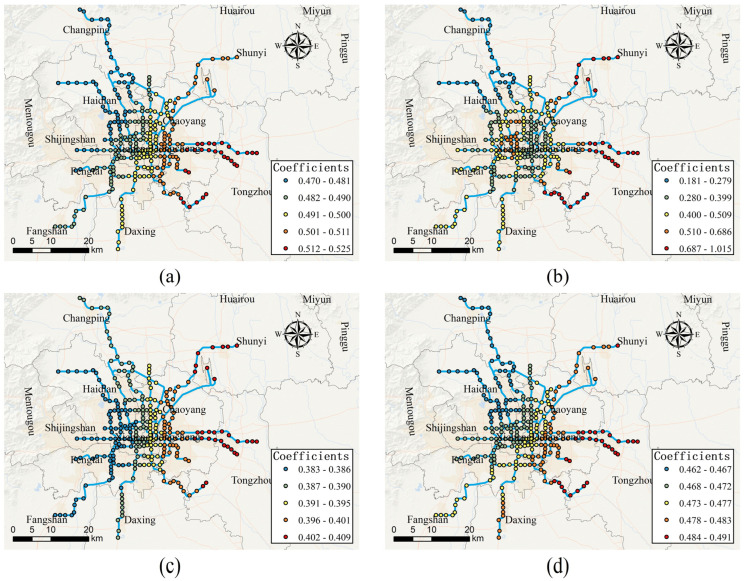
Spatially varying effects of hotels-residences POI density. (**a**) Effect of hotels-residences POI density on access-daily use on weekdays; (**b**) effect of hotels-residences POI density on egress-daily use on weekdays; (**c**) effect of hotels-residences POI density on access-daily use on weekends; (**d**) effect of hotels-residences POI density on egress-daily use on weekends.

**Figure 7 ijerph-19-02323-f007:**
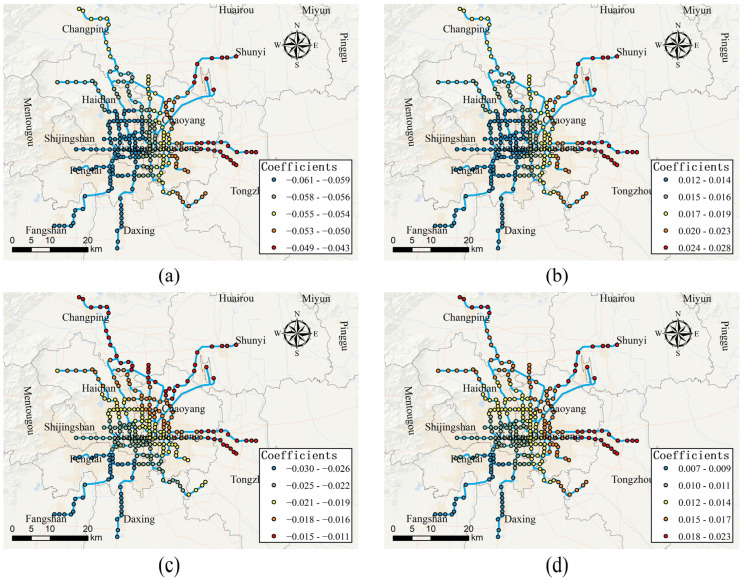
Spatially varying effects of primary road density. (**a**) Effect of primary road density on access-daily use on weekdays; (**b**) effect of primary road density on egress-daily use on weekdays; (**c**) effect of primary road density on access-daily use on weekends; (**d**) effect of primary road density on egress-daily use on weekends.

**Figure 8 ijerph-19-02323-f008:**
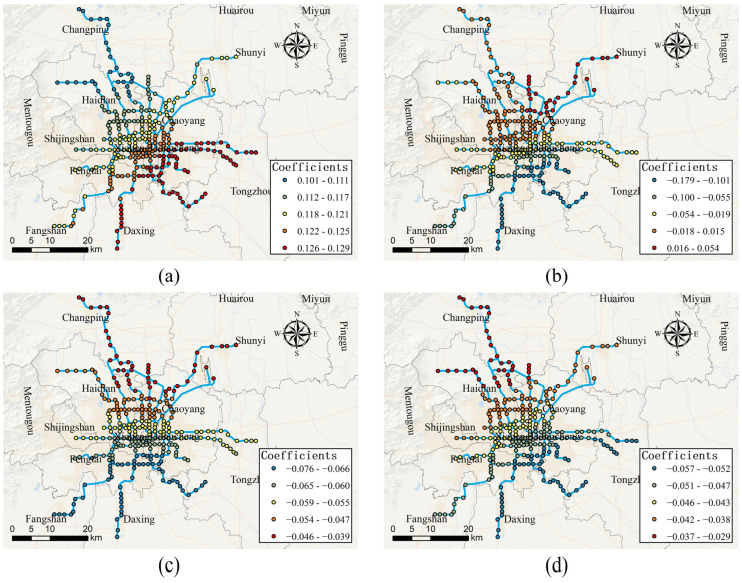
Spatially varying effects of branch road density and secondary road density. (**a**) Effect of branch road density on egress-daily use on weekdays; (**b**) effect of secondary road density on access-daily use on weekdays; (**c**) effect of secondary road density on access-daily use on weekends; (**d**) effect of secondary road density on egress-daily use on weekends.

**Figure 9 ijerph-19-02323-f009:**
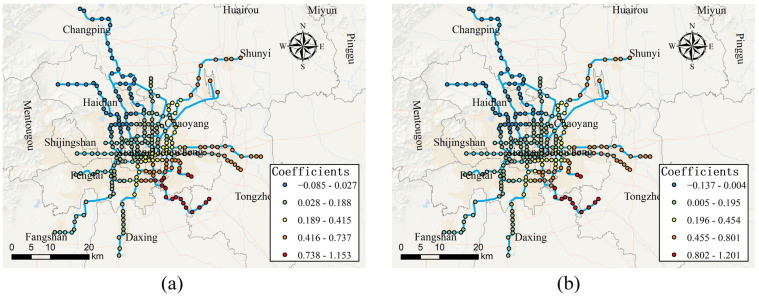
Spatially varying effects of population density. (**a**) Effect of population density on access-daily use on weekends; (**b**) effect of population density on egress-daily use on weekends.

**Table 1 ijerph-19-02323-t001:** Definition and summary statistics of the variables.

Categories	Variables	Abbreviation	Mean	Std.
Dependent variable	Access-daily use on weekdays (number per day)	/	162.810	133.663
	Egress-daily use on weekdays (number per day)	/	156.356	126.934
	Access-daily use on weekends (number per day)	/	106.585	92.015
	Egress-daily use on weekends (number per day)	/	104.602	89.301
Density	Population density (number/km^2^)	PopD	17.459	14.102
Diversity	Land-use mixture	LMix	0.921	0.073
Design	Primary road density (km/km^2^)	PriD	0.541	0.806
	Secondary road density (km/km^2^)	SecD	1.149	1.066
	Branch road density (km/km^2^)	BraD	2.702	1.428
Destination accessibility	Distance to CBD (km)	DCBD	13.609	8.526
Distance to transit	Bus station density (number/km^2^)	BusD	4.598	2.796
Other POIs	Restaurants-Shopping POI density (number/km^2^)	RSPD	62.424	52.904
	Companies-Banks POI density (number/km^2^)	CBPD	139.600	155.524
	Hotels-Residences POI density (number/km^2^)	HRPD	32.842	24.938
	Schools-Sports POI density (number/km^2^)	SSPD	76.039	65.748

**Table 2 ijerph-19-02323-t002:** Results of stepwise linear regression models.

Variables	Access-Daily Use on Weekdays		Egress-Daily Use on Weekdays		Access-Daily Use on Weekends		Egress-Daily Use on Weekends	
	p	VIF	p	VIF	p	VIF	p	VIF
DCBD	$0.000{\;}^{***}$	1.605	$0.000{\;}^{***}$	1.622	$0.000{\;}^{***}$	1.659	$0.000^{\;***}$	1.659
HRPD	$0.000{\;}^{***}$	1.858	$0.010{\;}^{***}$	1.838	$0.049{\;}^{**}$	2.053	$0.012{\;}^{**}$	2.053
PriD	$0.001{\;}^{***}$	1.130	$0.072{\;}^{***}$	1.069	$0.006^{\;***}$	1.139	$0.027{\;}^{**}$	1.139
SecD	$0.056{\;}^{*}$	1.192	-	-	$0.019{\;}^{**}$	1.196	$0.023{\;}^{**}$	1.196
BraD	-	-	$0.020{\;}^{**}$	1.267	-	-	-	-
PopD	-	-	-	-	$0.030^{\;**}$	1.438	$0.065^{\;*}$	1.438

Note: “***” p≤0.01; “**” p≤0.05; “*” p≤0.1; “-” means that it is not significant.

**Table 3 ijerph-19-02323-t003:** Results of Moran’s I test.

Variable	Moran’s I	z-Score	*p*-Value
DCBD	0.979444	26.639919	0.000
HRPD	0.615647	16.755601	0.000
PriD	0.523332	14.35121	0.000
SecD	0.428045	11.700967	0.000
BraD	0.310168	8.495692	0.000
PopD	0.412146	11.393449	0.000

**Table 4 ijerph-19-02323-t004:** Evaluation results of the OLS model, the GWR model, and the MGWR model.

**Metrics**	**Access-Daily Use on Weekdays**	**Egress-Daily Use on Weekdays**
**OLS**	**GWR**	**MGWR**	**OLS**	**GWR**	**MGWR**
$R^{2}$	0.240	0.485	0.498	0.256	0.570	0.559
$Adj.\;R^{2}$	0.229	0.398	0.442	0.246	0.480	0.500
AICc	750.483	726.361	686.154	744.359	697.108	660.930
Moran’s I(residuals)	0.258737(0.00) *	0.046537(0.17)	0.030440(0.35)	0.321465(0.00) *	0.058202(0.09)	0.051092(0.13)
**Metrics**	**Access-Daily Use on Weekends**	**Egress-Daily Use on Weekends**
**OLS**	**GWR**	**MGWR**	**OLS**	**GWR**	**MGWR**
$R^{2}$	0.193	0.510	0.509	0.215	0.552	0.546
$Adj.\;R^{2}$	0.179	0.408	0.446	0.201	0.451	0.481
AICc	769.942	734.265	689.091	762.113	718.003	674.457
Moran’s I(residuals)	0.260169(0.00) *	0.039950(0.22)	0.029660(0.35)	0.297075(0.00) *	0.042165(0.21)	0.030656(0.34)

Note: “*” p≤0.05

## Data Availability

Not applicable.

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
