# Peer review of "Exploring the Multiscale Relationship between the Built Environment and the Metro-Oriented Dockless Bike-Sharing Usage"

_ijerph, 2022, doi:10.3390/ijerph19042323_

Round 1
Reviewer 1 Report
Title: Exploring the multiscale relationship between the built environment and the metro-oriented dockless bike-sharing usage
Abstract
- Authors mentioned: “we established a multiscale geographic weighted regression (MGWR) model..” without mentioning OLS and GWR. But in the results, the authors compared the three methods.
- Please delete this sentence from the abstract: “However, few efforts have attempted to discuss the spatial relationship between the urban built environment and the integrated usage of bike-sharing and the metro, especially from the perspective of multiscale spatial relationship.”
- Mention the abbreviation of POI
Introduction
- Replace: “more and more popular” with “more popular”
- Please mentioned some studies/references on this issue: :The ordinary least square model is usually applied as the global model to explore the relationship between the dockless bike-sharing and the built environment,”
- “… is neglected by the OLS model.” It should be “is neglected by the related studies using OLS model.”
- Mention the abbreviation first, before authors mentioned OLS, GWR, etc.
Literature review
- Mention the reasons? Why is being neglected? “Although the above-mentioned regression model can capture the built environment factors influencing bike-sharing usage, the spatial non-stationarity of variables is neglected by them.”
- Mention their study results: “Wang, et al. [36] applied the GWR model to explore the relationship….”
- Replace: “the traditional GWR model ignores the differences between the impact scales” with “studies applying the traditional GWR…”
- Authors mentioned: “Previous studies demonstrated that the MGWR model is more reliable than the GWR model in dealing with the spatial relationship regression issue[20,21,39]”. Please explain the reasons.
Methodology
- Replace “methodology” with “method”
- Mention the abbreviation of “5DS”, “VIF”
- Equation 1 to 5, mention all notations meaning included in the equation. For example: E[I] and V[I] represent …
Result and Analysis
- This is my major concern, the authors only explain their model results without comparing with other studies elsewhere. It becomes hard to understand the academic contribution of this study. Please also explain what is your study contribution to the existing related studies.
Conclusion and Prospect
- Mention on each your policy recommendation, it refers to which findings? For example: the authors wrote: “Firstly, the bike-sharing companies can optimize the configuration of bicycles depending on the built environment.” So please add: “ This is because my study findings show that ….
- Do a similar thing on second and third recommendations.
- Related to your study limitation excluding socioeconomic factors. Why did authors neglect these factors. What are the reasons. Because in literature reviews section, the authors mentioned that previous studies found a correlation between socioeconomic and bike-sharing use.
Reviewer 2 Report
This is an excellent work that deserves to be published in the IJERPH.
The thematic is very pertinent and the results add new light to planning policies support.
I would only suggest a minor revision on the methodology and results analysis. There is no mention about topography. This is a strategic issue in many cities, as it determines also the configuration of metro lines. The relationship between topography, metro and dockless bike-sharing is a issue missing in this paper, that requests greater sensibility. Please add some info about it and add it to your conclusions.
Reviewer 3 Report
The present study employed sophisticated methods to establish multiscale regression model bike-sharing usage. This model present relevant environmental variables that affect bike-sharing usage.
In the introduction you mention studies that were conducted mainly in China or Japan. It would be useful for European reader also mention studies that were conducted in different cultures (Of course, if any studies on this topic exist?)
Minor point: Many abbreviations limit the readability of the text. Moreover, some abbreviations are not explained.
Abstract, line 19 What do „CBD“ and „POI“ mean?
The abbreviations MGWR, GWR, OLS are explained in the abstract, but should be also explained in the introduction.
Line 156 „METRO, WORK, LIVING, REC, MIX and POP“ – please explain, what do these abbreviations means.
Line 321 „The multiscale geographically weighted regression (MGWR) model“ – You describe the abbreviation with the full text, which is O.K., but it would be useful to explain the abbreviations more frequently in particular subsections.
Given that the length of the text is not limited in open access journals, I consider whether it is necessary to use such a large number of abbreviations.
Round 2
Reviewer 1 Report
I appreciate your efforts in revising the manuscript according to my suggestions. Your manuscript deserves to be published in this journal.
Best,
Reviewer